# Acrylamide and 5-Hydroxymethylfurfural in Synthetic Sugar Cane Syrup: Mitigation by Additives

**DOI:** 10.3390/molecules28073212

**Published:** 2023-04-04

**Authors:** Nuchnicha Phaeon, Pisittinee Chapanya, Anutin Pattamasuwan, Hanán Issa-Issa, Leontina Lipan, Ángel Antonio Carbonell-Barrachina, Esther Sendra, Klanarong Sriroth, Tanat Uan-on, Nuttakan Nitayapat

**Affiliations:** 1Department of Biotechnology, Faculty of Agro-Industry, Kasetsart University, 50 Ngamwongwan Rd., Lat Yao, Chatuchak, Bangkok 10900, Thailand; nuchnichap@mitrphol.com (N.P.); fagitnu@ku.ac.th (T.U.-o.); 2Mitr Phol Sugarcane Research Center Co., Ltd., 399 Moo 1 Chumpae-Phukieo Road, Khoksaat, Phukieo, Chaiyaphum 36110, Thailand; pisittineec@mitrphol.com (P.C.); anutinp@mitrphol.com (A.P.); klanarongs@mitrphol.com (K.S.); 3Research Group “Food Quality and Safety”, Centro de Investigación e Innovación Agroalimentaria y Agroambiental (CIAGRO-UMH), Department of Agro-Food Technology, Escuela Politécnica Superior de Orihuela, Universidad Miguel Hernández de Elche, Carretera de Beniel, km. 3.2, 03312 Orihuela, Alicante, Spain; hissa@umh.es (H.I.-I.); leontina.lipan@goumh.umh.es (L.L.); angel.carbonell@umh.es (Á.A.C.-B.); esther.sendra@umh.es (E.S.); 4Fruit Production Program, IRTA Mas Bové, Ctra. Reus-El Morell km. 3.8, 43120 Constantí, Tarragona, Spain

**Keywords:** calcium chloride, citric acid, HMF, interactions, non-centrifugal sugars, vitamin B3

## Abstract

The ability of additives to reduce the formation of acrylamide in simulated sugar cane syrups was investigated. Organic acids, B vitamins, and inorganic salts were added individually and in combination to simulated thickened cane juice, and the mixtures were heated at 120 °C for 30 min. Calcium chloride (1%), citric acid (0.1%), and vitamin B3 (0.1%) were the most effective individual additives from each chemical family. The effects of CaCl_2_ (0–1%), citric acid (0–0.125%), and vitamin B3 (0–0.1125%), when added in combination, on the concentrations of acrylamide and hydroxymethylfurfural (HMF) were studied using a Box–Behnken design. Combinations of all three additives lowered the acrylamide production, but only the combination of citric acid and vitamin B3 had a significant synergistic effect. However, all these additives stimulated the production of HMF, and no significant interactive effect between pairs of additives on HMF production was observed. Calcium chloride stimulated the formation of HMF most strongly. These results indicate that certain combinations of these additives effectively reduce acrylamide formation, but they also lead to an increase in the formation of HMF in sugar syrup.

## 1. Introduction

Potentially carcinogenic substances, such as acrylamide and 5-hydroxymethylfurfural (HMF), are produced during the cooking of foodstuffs when temperatures are elevated. Both acrylamide and HMF have been found in many cooked foods including deep-fried potatoes, roasted coffee, and non-centrifugal sugar (NCS) [1,2,3,4]. Acrylamide is formed in foods prepared by heating when their ingredients contain reducing sugars and amino acids. At temperatures higher than 100 °C, the Maillard reaction takes place and acrylamide is formed; however, under acidic conditions the formation of acrylamide is inhibited [5,6]. During caramelization reactions, the dehydration of reducing sugars at high temperatures produces HMF [7]. HMF is an intermediate of the Maillard reaction sequence and is formed by the dehydration of the Amadori rearrangement product, 1-amino-1-deoxy-2-ketose. Its formation is promoted by heating under acidic conditions [8].

In previous studies, the production of acrylamide during the manufacture of NCS that involved the evaporation of cane juice at high temperatures over an extended period was investigated, and concentrations of acrylamide of up to 4 mg kg^−1^ were found [3,9,10,11]. HMF has been found in sugar syrups, especially those containing high concentrations of fructose, such as honey [12]. This substance is also found in panela (an NCS manufactured in Central and South America) which is prepared by evaporating syrups rich in sucrose, and its content of HMF can be as high as 14.8 mg kg^−1^ [9,11].

Several methods describing the mitigation of acrylamide and HMF formation during the preparation of food have been previously reported [2,13]. The effects of additives, including amino acids, simple organic acids, inorganic salts, and water-soluble vitamins, on the concentrations of acrylamide and HMF found in foods or synthetic food models heated at high temperatures have been published, and some of these substances can inhibit the formation of these potential carcinogens by various mechanisms [14,15,16,17,18]. However, most of these studies only evaluated the use of single compounds. Combinations of additives might achieve the simultaneous inhibition of the formation of acrylamide and HMF so as to avoid the creation of unpleasant organoleptic characteristics caused by the addition of higher concentrations of single inhibitors that may even exceed their maximum legal thresholds. Since organic acids, salts, and B vitamins have been reported to reduce the formation of acrylamide in other heated foods, the aim of this study was to investigate the impact of such substances, both individually and in combination, on the extents of acrylamide and HMF formation in a sugar syrup produced by prolonged heating of artificial cane juice.

## 2. Results and Discussion

### 2.1. Effects of Various Single Acids, Salts, and B Vitamins on the Formation of Acrylamide

After heating, all of the additives apart from KCl and NaCl at the concentrations used caused more than 50% reductions in the content of acrylamide in the syrups. The content of acrylamide in the control (no additives) was 7.89 ± 0.2 mg kg^−1^ (Figure 1). Within each group of additives:The addition of 1% citric acid significantly reduced the acrylamide content of the syrup more effectively than 1% acetic acid (*p* < 0.05);Vitamin B3 was significantly more effective than the other two B vitamins at concentrations of 0.1% (*p* < 0.05);NaCl and KCl at concentrations of 1% were the least effective of all the additives examined while 1% CaCl_2_ was the most effective.

Thus, citric acid, vitamin B3, and CaCl_2_ were selected for the next step of the study, which was intended to investigate the effects of adding combinations of these three compounds on the concentrations of acrylamide and HMF produced.

Various studies have demonstrated that water-soluble vitamins are more effective in preventing acrylamide formation than fat-soluble vitamins [15]. Notably, B2, B5, and B6 have been identified as the most potent inhibitors among the B vitamins [19]. However, López-López, et al. [16] noted that B1, B3, and B6 at concentrations of 25–100 mM reduced up to 49% of acrylamide formation in heated ripe olive juice, although only B1 caused a statistically significant reduction. The present study employed concentrations of B vitamins of 3.0–8.2 mM, which resulted in a 60–85% decrease in acrylamide formation (Figure 1), and these concentrations are far lower than those used by López-López, et al. [16]. These findings imply that the composition of the food matrix is a critical factor influencing the effectiveness of B vitamins in inhibiting acrylamide formation.

Multivalent cations have been shown to inhibit acrylamide formation [20,21], and bivalent cations were more effective inhibitors than univalent species [22]. These findings agree with the results obtained in the current study. The greatest reduction in acrylamide was observed when 90 mM CaCl_2_ (1% CaCl_2_) was added, in comparison with the addition of 105 mM MgCl_2_ (1%), 134 mM KCl (1%), and 171 mM NaCl (1%) (Figure 1).

During the heating/evaporation stage of NCS production, the pH of the dilute juices decreases slowly as acidic compounds are formed by the degradation of monosaccharides [23]. This phenomenon was also observed in the current study (Figure 2); after heating at 120 °C for 30 min, the pH values of the syrups formed were significantly lower than those of the thick juices prior to the heating step (*p* < 0.05). Jung, et al. [24] have suggested that acidic pH values might decrease the formation of acrylamide by converting nucleophilic amino groupings into their protonated forms which do not react as nucleophiles at the Schiff base formation step of acrylamide production.

However, acidic pH also stimulates the hydrolysis of sucrose. The hydrolytic products (glucose and fructose) are reactants of the Maillard reaction. The total concentration of monosaccharides in the syrups produced when MgCl_2_ and CaCl_2_ were added was 45–55%, while this concentration was about 15% in syrups prepared in the absence of additives (Figure 3). The thick juice was prepared using a buffer solution of pH 8 in order to minimize the acid hydrolysis of sucrose. In the presence of some additives, the buffering capacity of this juice was not sufficient to maintain a slightly alkaline pH prior to exposure to 120 °C for 30 min, especially when salts with bivalent cations were added (Figure 2).

### 2.2. The Effects of Combinations of Additives on the Formation of Acrylamide and HMF

The addition of combinations of multiple additives might have a significant impact on the generation of acrylamide and HMF. Citric acid, vitamin B3, and CaCl_2_ were selected based on their performance in the determinations described above to investigate their interactions when added to thick juices in combination prior to heating at 120 °C for 30 min. The compositions of the combinations of additives investigated in each experiment are shown in Table 1.

In these experiments, the lowest concentration of acrylamide formed (1.33 ± 0.22 mg kg^−1^) was observed when 0.5, 0.125, and 0.1125% CaCl_2_, citric acid, and B3, respectively, were added (experiment number 12, Figure 4). However, this combination of additives resulted in the formation of nearly 6500 mg kg^−1^ of HMF, which was the highest concentration found after heating all of these juices. The lowest concentration of HMF produced (56.5 mg kg^−1^) was found when 0.0562% B3 was added (experiment number 1, Figure 4), but under these conditions 6.29 mg kg^−1^ of acrylamide was formed. From these experimental results, it can be stated that there was not a “simple” relationship between the formation of acrylamide and HMF.

In this study, the interactive effects of combinations of citric acid, vitamin B3, and CaCl_2_ on the concentrations of acrylamide and HMF produced by heating thick simulated cane juice at 120 °C for 30 min were investigated using response surface methodology (RSM). The results of the experiments presented in Figure 4 were subjected to multiple regression and response surface analysis. Equation (1) was derived to evaluate the effects of the three additives (CaCl_2_ (A), citric acid (B), and vitamin B3 (C)) on the concentration of acrylamide formed (Y_A_), and the results of the ANOVA analysis of the fits are shown in Table 2.
Y_A_ = 6.24147 − 2.13322(A) − 0.66874(B) + 8.31767(C) + 1.34(AB) + 5.94524(AC) − 155.23627(BC) − 0.47361(A^2^) − 30.17248(B^2^) − 179.23516(C^2^)(1)

The *p*-values for the cross factors for combinations containing CaCl_2_ and citric acid (A-B) and CaCl_2_ and vitamin B3 (AC) shown in Table 2 indicated that there was no significant interaction between these pairs of additives. However, the *p*-value of the cross factor for combinations of citric acid and vitamin B3 (BC) indicated that there was a significant interaction between this pair. The response surfaces and contour plots constructed using Equation (1) (illustrating changes in acrylamide formation when concentrations of the additives were varied) are shown in Figure 5.

These plots show that all of the combinations of additives decreased the concentration of acrylamide produced. The absence of significant interactions of combinations of CaCl_2_ and citric acid and CaCl_2_ and vitamin B3 is shown by the parallel edges of the response surface in Figure 5a,c. These results indicate that there is a significant interactive effect on the formation of acrylamide when combinations of citric acid and vitamin B3 are used, as shown in Figure 5e. Increasing the concentrations of citric acid and vitamin B3 in combination significantly reduces the formation of acrylamide compared with increasing the concentrations of these additives when used individually.

Equation (2) was built to evaluate the effects of three additives (CaCl_2_ (A), citric acid (B), and vitamin B3 (C)) on the concentration of HMF formed (Y_H_), and the results of the ANOVA analysis of the fits are shown in Table 3. In order to obtain good fits to the experimental data, the elimination of the term BC was required, indicating that there was no significant interactive effect between citric acid and vitamin B3 on the formation of HMF. Moreover, the *p*-values of the cross factors AB and AC (Table 3) and the response surface plots (Figure 6a,c) showed that neither citric acid nor vitamin B3 had any significant interactive effect with CaCl_2_ on the concentration of HMF produced. Figure 6 also clearly illustrates that all of the additives stimulated the production of HMF, and CaCl_2_ was the additive that had the strongest effect. Increasing the concentration of CaCl_2_ caused more HMF to be produced than by increasing the concentrations of the other additives. The increased HMF generation caused by CaCl_2_ observed here agrees with results reported earlier using other food matrices [22,25].
Y_H_ = −592.45611 + 12,476.03194(A) + 16,724.33556(B) + 21,907.46173(C) + 6186.50667(AB) + 16,064.32593(AC) − 9133.68444(A^2^) − 82,819.69778(B^2^) − 1.39103 × 10^5^(C^2^)(2)

When 1% CaCl_2_ was added, the syrup produced (pH 3.79) was the most acidic when compared with the syrups produced in the presence of the other additives (Figure 2), and the greatest extent of hydrolysis of sucrose was observed in this syrup (Figure 3).

HMF is formed by heating under acidic conditions [26], and it appears to be formed in all materials containing monosaccharides which are heated [27]. These findings might explain the increase in concentrations of HMF formed when the concentration of CaCl_2_ was increased (Figure 6).

Gökmen and Şenyuva [22] reported that cations reduced the formation of acrylamide but increased HMF formation during heating of a glucose–asparagine model system. They suggested that the cations diverted the reaction path toward the dehydration of glucose leading to the formation of HMF and furfural due to their interference with Schiff base formation which normally leads to the production of acrylamide. Yang, et al. [25] also noted that calcium ions significantly enhanced HMF formation when its concentration was increased from 10 to 80 ppm in honey heated at 90 °C for 4 h.

## 3. Materials and Methods

### 3.1. Materials

Acrylamide (>99.8%), asparagine, glutamine, aspartic acid, alanine, valine, gamma-aminobutyric acid (GABA), NaCl, KCl, MgCl_2_, CaCl_2_, thiamine hydrochloride (vitamin B1), nicotinic acid (vitamin B3), pyridoxal hydrochloride (vitamin B6), sucrose, D-glucose, D-fructose, glacial acetic acid (≥99%), and citric acid were obtained from Sigma-Aldrich (St. Louis, MO, USA). Analytical-standard-grade (97%) 5-hydroxymethylfurfural (HMF) was purchased from Thermo Fisher (Kandel GmbH, Kandel, Germany).

### 3.2. Preparation of Synthetic Thick Cane Juice

Simulated cane juice (1 kg) was prepared by dissolving the following components in 0.2 M KH_2_PO_4_-NaOH buffer (pH 8): sucrose (568.75 g), glucose (58.68 g), fructose (48.75 g), asparagine (1.461 g), glutamine (0.0116 g), aspartic acid (0.017 g), alanine (0.015 g), valine (0.017 g), GABA (0.017 g), KCl (1.72 g), CaCl_2_ (0.53 g), MgCl_2_ (1.97 g), and NaCl (0.64 g). The juice was heated at 65–75 °C for about 2 h to achieve a thickened juice with a total solid concentration of 65.0 ± 0.5° Brix [3]. The thick juice was stored at 4 °C.

### 3.3. The Effects of Additives on Acrylamide Formation

To the thick cane juice, the following additives were added: (i) acetic acid or citric acid (0.12% *w*/*w*, final concentration); (ii) B vitamins (0.1% *w*/*w*); (iii) inorganic salts (1% *w*/*w*); or (iv) combinations of these compounds at various concentrations, which will be described in detail in the following section. The mixtures were heated at 120 °C for 30 min to mimic the process of syrup production. Thick cane juice without additives served as a control.

### 3.4. Analyzing the Effects of Combined Additives: Response Surface Methodology

A response surface methodology involving Box–Behnken design (BBD) was applied to design the experiments to investigate the effects of several substances added to thick juice simultaneously. The individual additives (CaCl_2_ (X_1_), citric acid (X_2_), and vitamin B3 (X_3_)) which caused the greatest reduction in acrylamide concentrations within each group of additives (inorganic salts, organic acids, and B vitamins) were selected to investigate their effects in combination on the concentration of acrylamide and HMF in the syrup produced by heating at 120 °C for 30 min. Three levels (low, medium, and high), coding levels, and codes of these additives are shown in Table 4. The highest concentrations of CaCl_2_ (1.0% *w*/*w*), citric acid (0.125% *w*/*w*), and vitamin B3 (0.1125% *w*/*w*) were selected based on their effects on the unpleasant taste which could be detected by sensory analysis, the pH of syrup in which a pH lower than 4.5 would significantly stimulate the acid hydrolysis of sucrose, and the recommended maximum daily intakes quoted in relevant international standards and by health agencies.

The 15 experiments defined by the BBD (performed in triplicate) included 3 replicates performed at the center point (Table 1). Design-Expert 8.0.7 software was used to analyze the experimental data in order to derive a full quadratic-form model (Equation (3)) that describes the fit to these data:*Y* = *B*_0_ + *b*_1_*X*_1_ + *b*_2_*X*_2_ + *b*_3_*X*_3_ +*b*_12_*X*_1_*X*_2_ + *b*_13_*X*_1_*X*_3_ + *b*_23_*X*_2_*X*_3_ + *b*_11_*X*_1_^2^ +*b*_22_*X*_2_^2^ + *b*_33_*X*_3_^2^(3)
where *Y* is the predicted concentration of acrylamide or HMF (mg kg^−1^) produced; *B*_0_ is a constant; *X*_1_, *X*_2_, and *X*_3_ are the concentrations of CaCl_2_, citric acid, and vitamin B3 (%), respectively; *b*_1_, *b*_2_, and *b*_3_ are the coefficients of the linear terms; *b*_12_, *b*_13_, and *b*_23_ are the coefficients of the cross additives; and *b*_11_, *b*_22_, and *b*_33_ are the coefficients of the quadratic terms.

The software also provided the determination coefficient (R^2^), adjusted determination coefficient (adjusted R^2^), predicted determination coefficient (predicted R^2^), and lack of fit error, which were used to assess the quality of the fits to the experimental data. The F- and *p*-values from the analysis of variance (ANOVA) were used to test the significance of the effects of additives.

### 3.5. Analytical Methods

The contents of acrylamide and sugars were determined as described previously [3], while HMF content was determined by the method described by Vázquez Araújo, et al. [28]. Syrup (approx. 10 g) was dissolved in 25 mL of ultrapure water, and the solution was filtered (0.45 μm filter). After dilution with water, portions (20 µL) were applied to an HPLC (Hewlett Packard model 1100 Series, Wilmington, DE, USA) using a stainless-steel LiChroCART^®^ 250–4, 5µm HPLC cartridge packed with LiChrospher^®^ RP-8, and elution at a flow rate of 1 mL min^−1^ was used with a mobile phase containing acetonitrile/water (6/94 *v*/*v*). Detection of HMF was conducted using a diode array detector (Hewlett Packard model 1100 Series, Wilmington, DE, USA) set to 285 nm. HMF was quantified using an external calibration curve constructed using analytical-grade HMF. All determinations (sugars, acrylamide, and HMF) were carried out in triplicate.

### 3.6. Statistical Analysis

One-way analysis of variance and Tukey’s multiple range test were used to check the difference among the effects of the individual additives; *p* < 0.05 was considered to be statistically significant. Calculations were performed using XLSTAT Premium 2016 software (Addinsoft, New York, NY, USA).

## 4. Conclusions

The results show that a combination of additives can be used to address the limitations of using a single additive at higher concentrations, which may result in undesirable organoleptic characteristics in food products or exceeding the maximum daily intake recommended by health authorities. Although CaCl_2_ effectively inhibited acrylamide production in cane syrup, it stimulated HMF formation strongly. Only the combination of citric acid and vitamin B3 showed synergistic inhibitory effects on acrylamide formation and did not significantly increase the concentration of HMF formed. Therefore, a combination of citric acid and vitamin B3 can be employed to effectively mitigate the formation of acrylamide without increasing HMF formation in sugar cane syrup, and it is proposed that the addition of CaCl_2_ should be avoided. Further investigation of their mechanisms of inhibition is needed to understand the interactions between these additives and the chemical processes involved in the formation of acrylamide and HMF.

## Figures and Tables

**Figure 1 molecules-28-03212-f001:**
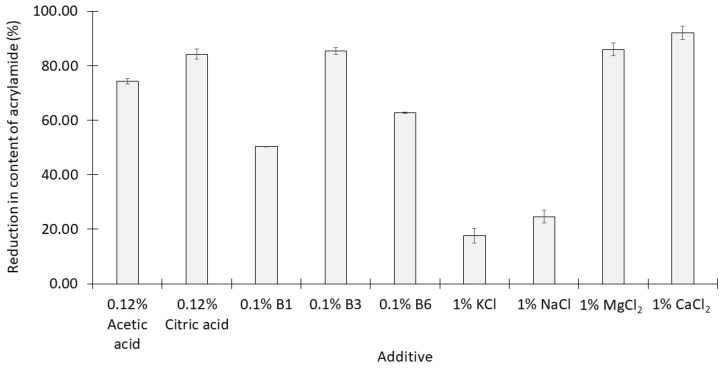
Reduction in the content of acrylamide (mean ± SD, *n* = 3) of syrup produced by single acids, vitamins, and salts added to thick simulated cane juice prior to heating at 120 °C for 30 min.

**Figure 2 molecules-28-03212-f002:**
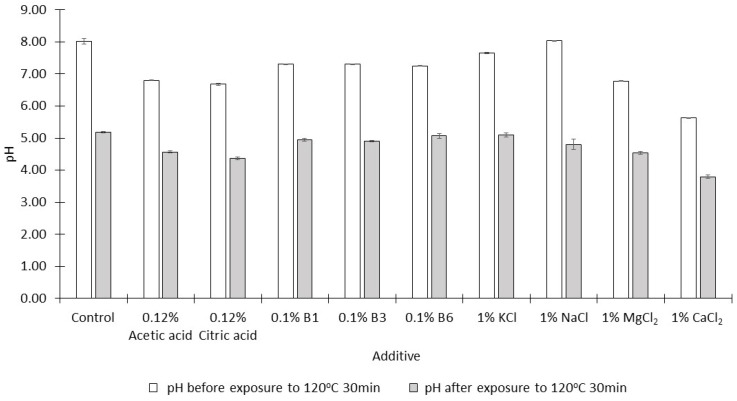
The pH of mixtures (mean ± SD, *n* = 3) containing single additives before and after heating at 120 °C for 30 min.

**Figure 3 molecules-28-03212-f003:**
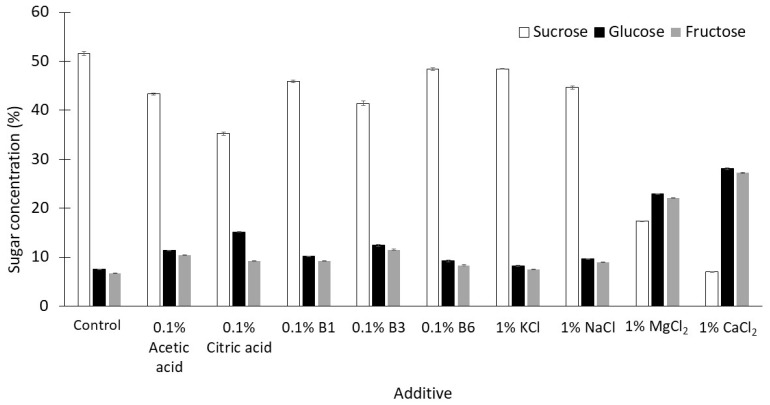
Concentrations of sugars (mean ± SD, *n* = 3) in syrups prepared from thick juices containing single additives by heating at 120 °C for 30 min.

**Figure 4 molecules-28-03212-f004:**
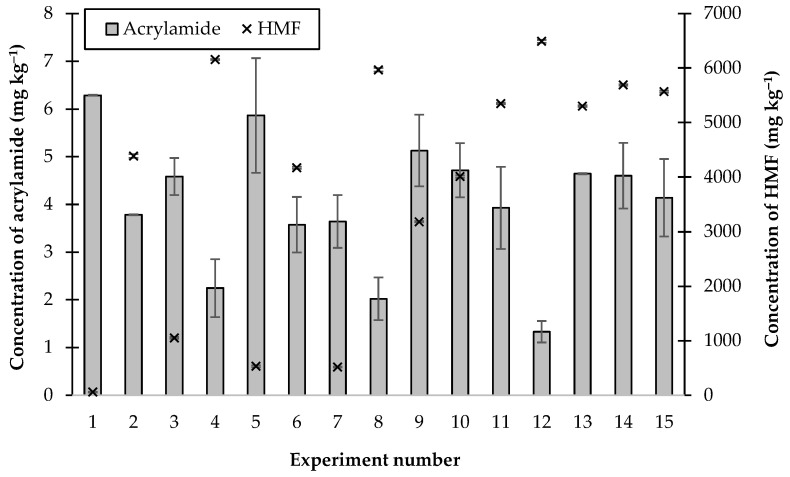
The concentrations of acrylamide and HMF produced by heating thick juice containing combinations of additives at 120 °C for 30 min.

**Figure 5 molecules-28-03212-f005:**
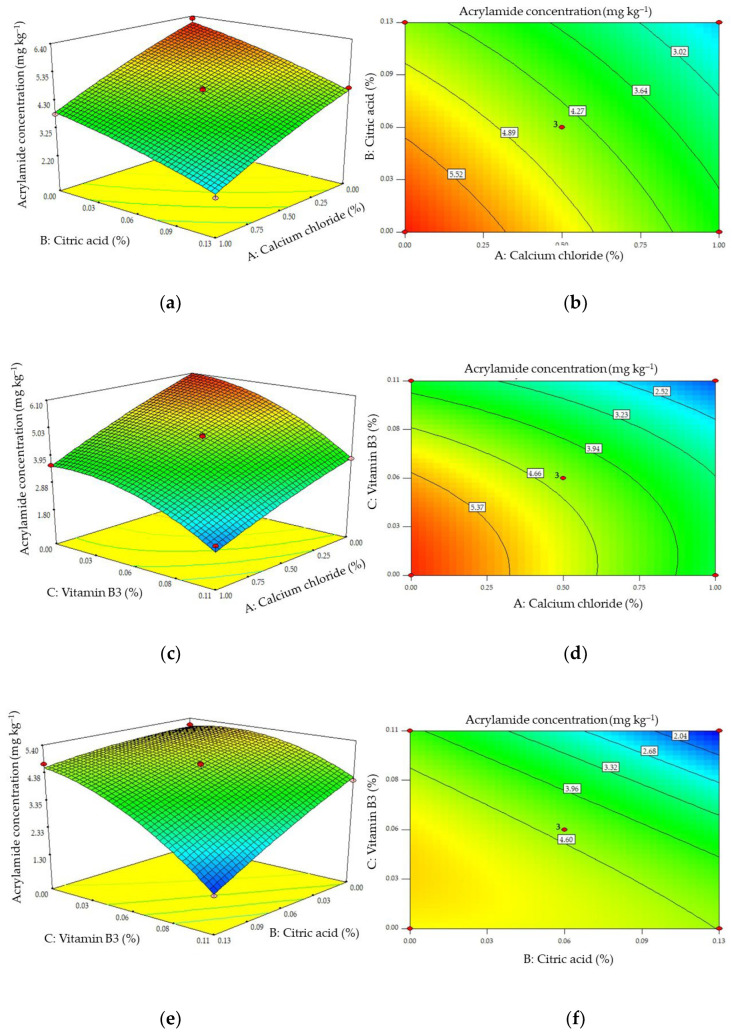
Response surface (**a**,**c**,**e**) and contour (**b**,**d**,**f**) plots of the effects of pairs of additives on the concentration of acrylamide formed with the concentration of the third additive at its mid-level.

**Figure 6 molecules-28-03212-f006:**
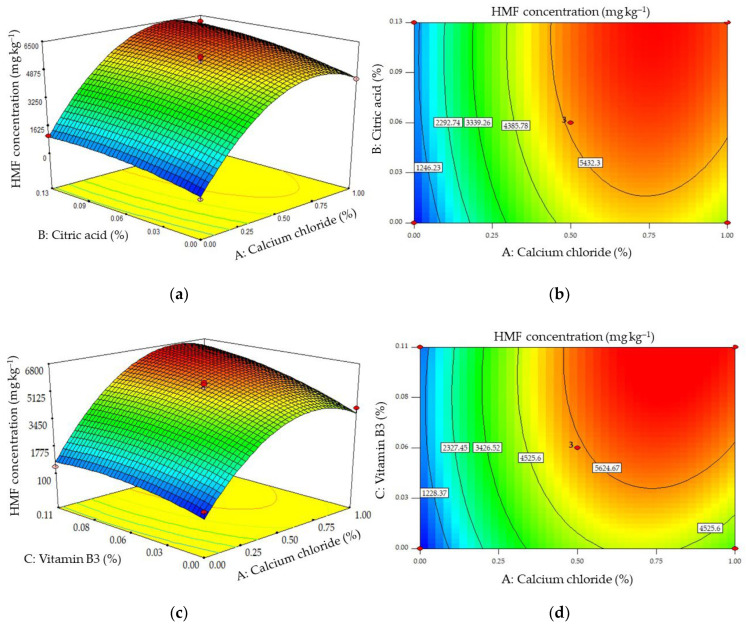
Response surface (**a**,**c**) and contour (**b**,**d**) plots of the effects of pairs of additives on the concentration of HMF formed.

**Table 1 molecules-28-03212-t001:** Box–Behnken design to evaluate combinations of additives affecting the formation of acrylamide and HMF in thick simulated cane juice heated to 120 °C for 30 min.

Experiment Number	Additive Factors		
	[X_1_ Coding Level] CaCl_2_ (% *w*/*w*)	[X_2_ Coding Level] Citric Acid (% *w*/*w*)	[X_3_ Coding Level] Vitamin B3 (%*w*/*w*)
1	[−1] 0	[−1] 0	[0] 0.0562
2	[1] 1.0000	[−1] 0	[0] 0.0562
3	[−1] 0	[1] 0.1250	[0] 0.0562
4	[1] 1.0000	[1] 0.1250	[0] 0.0562
5	[−1] 0	[0] 0.0625	[−1] 0
6	[1] 1.0000	[0] 0.0625	[−1] 0
7	[−1] 0	[0] 0.0625	[1] 0.1125
8	[1] 1.0000	[0] 0.0625	[1] 0.1125
9	[0] 0.5000	[−1] 0	[−1] 0
10	[0] 0.5000	[1] 0.1250	[−1] 0
11	[0] 0.5000	[−1] 0	[1] 0.1125
12	[0] 0.5000	[1] 0.1250	[1] 0.1125
13	[0] 0.5000	[0] 0.0625	[0] 0.0562
14	[0] 0.5000	[0] 0.0625	[0] 0.0562
15	[0] 0.5000	[0] 0.0625	[0] 0.0562

**Table 2 molecules-28-03212-t002:** Analysis of variance: Box–Behnken quadratic response surface model for acrylamide formation (Equation (1)).

Source	Sum of Squares	Degree of Freedom	Mean Square	F-Value	*p*-Value Prob > F
Model (Equation (1))	25.74	9	2.86	40.06	0.0004
A—CaCl_2_	9.58	1	9.58	134.22	<0.0001
B—citric acid	4.88	1	4.88	68.43	0.0004
C—vitamin B3	8.73	1	8.73	122.37	0.0001
AB	7.014 × 10^−3^	1	7.014 × 10^−3^	0.098	0.7666
AC	0.11	1	0.11	1.57	0.2660
BC	1.19	1	1.19	16.69	0.0095
A^2^	0.052	1	0.052	0.73	0.4333
B^2^	0.051	1	0.051	0.72	0.4353
C^2^	1.19	1	1.19	16.64	0.0096
Residual	0.36	5	0.071		
Lack of fit	0.20	3	0.067	0.85	0.5796
Pure error	0.16	2	0.078		
Cor total	26.09	14			

Standard deviation = 0.27, mean = 4.03, coefficient of variation = 6.63%, predicted residual sum of squares = 3.56, R^2^ = 0.9863, adjusted R^2^ = 0.9617, predicted R^2^ = 0.8637, and adequate precision = 21.721.

**Table 3 molecules-28-03212-t003:** Analysis of variance: Box–Behnken-modified quadratic response surface model of HMF formation (Equation (2)).

Source	Sum of Squares	Degree of Freedom	Mean Square	F-Value	*p*-Value Prob > F
Model (Equation (2))	7.143 × 10^7^	8	8.929 × 10^6^	43.77	<0.0001
A—CaCl_2_	4.292 × 10^7^	1	4.292 × 10^7^	210.39	<0.0001
B—citric acid	2.800 × 10^6^	1	2.800 × 10^6^	13.72	0.0100
C—vitamin B3	5.169 × 10^6^	1	5.169 × 10^6^	25.34	0.0024
AB	1.495 × 10^5^	1	1.495 × 10^5^	0.73	0.4248
AC	8.165 × 10^5^	1	8.165 × 10^5^	4.00	0.0924
A^2^	1.925 × 10^7^	1	1.925 × 10^7^	94.37	<0.0001
B^2^	3.864 × 10^5^	1	3.864 × 10^5^	1.89	0.2179
C^2^	7.153 × 10^5^	1	7.153 × 10^5^	3.51	0.1103
Residual	1.224 × 10^6^	6	2.040 × 10^5^		
Lack of fit	1.146 × 10^6^	4	2.864 × 10^5^	7.30	0.1241
Pure error	78,490.83	2	39,245.42		
Cor total	7.266 × 10^7^	14			

Standard deviation = 451.68, mean = 3896.09, coefficient of variation = 11.59%, predicted residual sum of squares = 1.157 × 10^7^, R^2^ = 0.9832, adjusted R^2^ = 0.9607, predicted R^2^ = 0.8407, and adequate precision = 17.836.

**Table 4 molecules-28-03212-t004:** Additives, their levels, and coding.

Additive	Code	Coding Level		
		−1	0	+1
CaCl_2_ (% *w*/*w*)	X_1_	0	0.5000	1.0000
Citric acid (% *w*/*w*)	X_2_	0	0.0625	0.1250
Vitamin B3 (% *w*/*w*)	X_3_	0	0.0562	0.1125

## Data Availability

The original data presented in the study are included in the article; further inquiries can be directed to the corresponding authors.

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
