# Peer review of "Acrylamide and 5-Hydroxymethylfurfural in Synthetic Sugar Cane Syrup: Mitigation by Additives"

_molecules, 2023, doi:10.3390/molecules28073212_

Round 1

Reviewer 1 Report

General comments for the authors

1.      line 20  HMF should mention the full name for the first time

2.       Line 25  CaCl2 should not start your sentence with an abbreviation, please

3.      Line 26 show  = showed

4.       Line 27  enhance = enhanced

5.      Line 33  at= should be when elevated temperatures

6.      Line 33 They  = should mention they refer to what

7.      Line 35 ( Acrylamide has been detected in foods where ingredients con- taining reducing sugars and amino acids are heated) this sentence should be rewrite again please

8.      Line 42 ( In an earlier study) and in the same sentence (was studied) so please try to reconstruct this sentence

9.      The hypotheses of the study should be emphasized clearly please

10.  Line 67 please remove as  

11.  Line 69 please remove treatment

12.  Line 70 should remove significantly when you already mentioned the p value , and the same for all the manuscript, please

13.  In my opinion for the authors when you get a significant effect you better emphasize the p value much better   

14.  Figure 2 and 3,  I suggest to put control instead of no additives

15.  Line 320 HPLC should mention the  manufacturing company, city and country as well the same for  the UV detector

16.  The conclusion is so long please try to shorten it, please

Author Response

The authors would like to thank the reviewer for these helpful comments and recommendations. We attach the pdf with the answers to each comment, in addition, in the last version of the manuscript, we have highlighted all the changes with the green color to be easier to identify.

Reviewer 2 Report

Firstly, the selection of the additives should be justified in the introduction section by explaining their potential role in the formation of acrylamide and HMF. This will help readers understand why these specific additives were chosen for the study.

Secondly, as you pointed out, the introduction section should be expanded to provide more detailed information on the mechanisms of HMF formation. This will help readers understand the importance of studying the effects of the additives on HMF formation.

Thirdly, the results and discussion section should be written in a more organized and coherent manner. The results should be presented in a clear and concise manner, and the discussion should provide a comprehensive interpretation of the results.

Finally, the conclusion section should be revised to reflect the limitations of the study and the need for further research to fully understand the potential interactions between the additives and the formation of acrylamide and HMF.

Overall, I agree that the research idea is great, and with some revisions, the article could be suitable for publication.

23-24 : There was no noticeable interactive effect between pairs of additives on the production of HMF.

It is recommended that the authors extensively revise their Results and Discussion section as it is currently poorly written. The few existing sentences that have been corrected (see below) provide some guidance on how to present data in a more scientific manner. However, a more comprehensive overhaul of the section is necessary to ensure clarity and accuracy in the reporting of findings.

26-28 : The results indicate that certain combinations of these additives can effectively reduce the formation of acrylamide, but they also lead to an increase in the formation of HMF in sugar syrup that is prepared by heating at high temperatures.

40-41 : HMF is a product of the Maillard reaction and its formation is promoted by cooking in acidic conditions

4é : A previous study investigated the production of acrylamide during the manufacturing of NCS by evaporating cane juice at high temperatures over an extended period.

59-61 : The purpose of this study was to investigate the impact of salts, organic acids, and B vitamins, both individually and in combination, on the level of acrylamide and HMF formation in a sugar syrup produced by evaporating artificial cane juice through prolonged heating.

76-84 : Various studies have demonstrated that water-soluble vitamins are more effective in preventing acrylamide formation than fat-soluble vitamins. Notably, B2, B5, and B6 have been identified as the most potent inhibitors among the B vitamins. However, in López-López et al.'s research, B1, B3, and B6 at concentrations of 25-100 mM reduced up to 49% of acrylamide formation in heated ripe olive juice, with only B1 displaying a significant reduction. The present study employed lower concentrations of B vitamins (3.0-8.2 mM), which resulted in a 60-85% decrease in acrylamide formation, far less than that used by López-López et al. These findings imply that the composition of the food matrix is a critical factor influencing the effectiveness of B vitamins in inhibiting acrylamide formation.

123-124 : The incorporation of multiple additives into juices can have a notable impact on the generation of acrylamide and HMF

Table 1 : Maitain the same format when reporting numbers « decimals »

147-150 : No need to give more details and definition on RSM, remove

BE CONSITENT / Past or present tense ?????

176_179 : poor writing : replace by : The results indicate that there is a significant interactive effect on the formation of acrylamide when combinations of citric acid and vitamin B3 are used, as demonstrated in Figure 5e. Increasing the concentrations of citric acid and vitamin B3 in combination significantly reduces the formation of acrylamide compared to increasing the concentrations of these additives when used individually.

333-335 ; replace by Using a combination of additives can help to address the limitations of using a single additive at higher concentrations, which may result in undesirable organoleptic characteristics in food products or exceed the maximum daily intake recommended by health authorities.

336-end : Poorly written, please re-write it

Author Response

The authors are grateful for the reviewer's advice and much helpful suggestion. Please find our answer point by point in the attached pdf. In addition, we have highlighted directly in the manuscript blue color all the changes to be easier to identify.
